# Parents Reaching Out to Parents: An Appreciative, Qualitative Evaluation of Stakeholder Experiences of the Parent Champions in the Community Project

**DOI:** 10.3390/children9101479

**Published:** 2022-09-27

**Authors:** Bernie Carter, Anita Flynn, Jacqueline McKenna

**Affiliations:** Faculty of Health, Social Care and Medicine, Edge Hill University, Ormskirk L39 4QP, UK

**Keywords:** bronchiolitis, parent champion, deprivation, hospitalization, peer-led support, socio-economic burden, health promotion, community, intervention, respiratory syncytial virus

## Abstract

Background: Bronchiolitis is a seasonal, global acute lower respiratory tract infection caused by respiratory syncytial virus (RSV) and is a leading cause of hospital admission in young children. A peer-led (parents to parents) intervention was implemented to empower parents of children at risk of bronchiolitis and reduce hospital admissions. This paper reported the evaluation that aimed to gain the perspectives and experiences of five key stakeholder groups. Methods: A qualitative remote interview-based design, informed by Appreciative Inquiry was used. Thematic analysis was used. Results: Sixty-five stakeholders participated: parents (n = 43; mothers, n = 42), Parent Champions (n = 9), Children’s Centre Managers (n = 8), Children’s Centre Group Leaders (n = 11), and Core Team (n = 4). An overarching theme ‘Parents reaching out to parents’ was supported by five sub-themes (Raising awareness and sharing knowledge; Creating connection, trust, and confidence; Flourishing in their role as a Parent Champion; Rising to the challenges; and Knowledge is power, prevention is key: the government needs to know this.) Conclusions: Parent-to-parent peer support via the Parent Champions was perceived positively by parents who wanted to learn and improve the lives and health of their children. Parent Champions were successful in delivering information. Considering the socioeconomic burden of bronchiolitis to services and families, the potential for an upstream, relatively low cost, high-reach innovative intervention, as evidenced in this project, seems a valuable opportunity for improving children’s respiratory health.

## 1. Introduction

Bronchiolitis is a seasonal, global acute lower respiratory tract infection caused by respiratory syncytial virus (RSV); worldwide 150 million new cases are reported annually [1]. Epidemics typically occur in the winter months [2] although out-of-season and other changes were noted during [3,4,5,6] and after [4] the main waves of the COVID-19 pandemic. Bronchiolitis is a leading cause of hospital admission in young children [2,7]; thus, creating an economic and other burden on both health services [7,8] and on families [9], particularly when children require intensive care [10].

There is increasing consideration of improving value in care for bronchiolitis by reducing the use of ineffective, wasteful, and costly interventions [11] but much less on reducing the number of children contracting bronchiolitis using non-pharmaceutical interventions [6], and public health and health promotion strategies. Vaccines are being developed [12,13] but, even if effective, they will not be a complete answer to prevention. Better education of health care professionals and public health measures such as reducing second-hand smoke exposure [14] and health promotion advice (e.g., hand washing, social distancing) that have been effective in inhibiting the spread of infectious disease during the COVID-19 pandemic [5,6] could contribute to reducing the spread and impact of bronchiolitis. There is limited evidence for the benefits of empowering parents through information to address knowledge deficits [15] prior to their child contracting bronchiolitis [16], during any subsequent hospitalisation [17] and as well as follow-up post hospitalisation [18]. However, there are recommendations that key safety information should be given to parents looking after their baby with bronchiolitis at home [19].

Socioeconomic deprivation [20,21,22,23], indoor pollution [1,24] such as second-hand smoke exposure [14,25,26] and inhaling cooking oil fumes [1], outdoor pollution [1,24], prematurity [22,27], low parental education [27], and low breastfeeding rates and/or duration of breastfeeding [27,28] are all reported as risk factors and associated with hospital admissions. Poverty is associated with worse outcomes for children who require critical care for bronchiolitis [23].

Bronchiolitis causes 1 in 6 of all United Kingdom (UK) winter hospitalisations of babies and children [29]; nearly half of these hospitalisations are in babies under 6 months old. In Scotland, the highest admissions have been reported in infants aged 1 month and children from the most deprived quintile [2]. In England admissions are positively associated with area-level deprivation [30] and, at city level, Liverpool (NW England, UK), the fourth most socio-economically deprived local authority in England, has bronchiolitis admission rates that are consistently twice the national average [31].

This paper reported the findings from the interviews undertaken as part of the evaluation of the Parent Champion in the Community (PCC) project (October 2021–March 2022). This is the first peer-led project specifically focusing on bronchiolitis.

This project aimed to provide peer-led support to families in the most deprived areas in Liverpool whose babies are most at risk of having severe bronchiolitis. This support included empowering and educating expectant/new parents to address severe bronchiolitis risk factors, make informed lifestyles choices and access help in enabling these, and educate parents in self-managing bronchiolitis and how and when to seek medical help. The PCC project was based in ten Children’s Centres in the most deprived parts of Liverpool. The Children’s Centres work with families from the antenatal period up to the age of 5 years, offering a programme of holistic family support.

The aim of the evaluation was to gain the perspectives (e.g., their thoughts and ideas about the project), and experiences (what actually happened) of the key stakeholders about the PCC project. The particular focus of the evaluation was on what difference the project made to them, and as relevant, to their family, their setting/organisation, and the community.

## 2. Materials and Methods

A qualitative interview-based design, informed by an Appreciative Inquiry (AI) perspective, was used. Appreciative Inquiry is an affirmative, collaborative, relational and democratic approach to undertaking research and evaluation [32,33,34,35] that is grounded in working with existing strengths and successes within a setting or an organisation and working collaboratively, energetically and co-operatively to build a better future [36]. Appreciative Inquiry is underpinned by generative and affirmative language [35,37]. This does not mean that researchers adopting an Appreciative Inquiry lack criticality in their approach, as negative situations are not disregarded but encompassed within discussions [38].

Ethics approval was gained via the Health Research Ethics Committee at Edge Hill University (ETH2122-0054).

Five sets of stakeholders were invited to participate in the interviews: (1) Parent Champions appointed as part of the PCC project and working in participating Children’s Centres; (2) parents who engaged with Parent Champions; (3) managers from the participating Children’s Centres; (4) group leaders running key groups/activities in the Children’s Centres; and (5) core implementation team members.

Each of these stakeholder groups had a specific role in the intervention and, thus, their perspectives were important to capture as part of the evaluation. The role of the Parent Champions was in directly delivering the intervention to the parents who were in direct receipt of the intervention. The role of the Children’s Centre Managers was to both manage the Parent Champions and strategically support the intervention. Children’s Centre Group Leaders invited the Parent Champions into their groups and activities (e.g., Breastfeeding Group, Stay and Play Group) so that the Parent Champions could engage with parents and deliver their information.

Purposive sampling was used for all stakeholder groups, apart from the parent group where convenience sampling was used. All Parent Champions, Children’s Centre Managers, and Children’s Centre Group Leaders working in the intervention settings were initially contacted by their line manager, who acted as a gatekeeper. Those who were interested in being interviewed agreed for their preferred contact address to be shared by their line manager with the evaluation team. The Parent Champions chose WhatsApp for initial contact and the other stakeholders were contacted by email.

Of the 322 parents who had participated in a survey in Phase 1 of the evaluation, 61 ticked a box in the survey to state their interest in taking part in an interview. They indicated they agreed for their contact details to be shared by their Parent Champion with the evaluation team. Contact was attempted (up to three times) with all 61 eligible parents (telephone/email) by a member of the evaluation team within two–three days of their contact with their Parent Champion.

Appreciatively oriented conversational interviews were carried out with all stakeholder groups by members of the evaluation team (October 2021–March 2022). Core questions, tailored to each stakeholder group, that reflected the aims of the evaluation and PCC project were asked, as relevant, across all stakeholder groups (see Appendix A). Questions for Parent Champions included what worked well in their training, challenges they faced, and what they thought their ‘superpowers’ were. Parent questions included what they found helpful, what they had learned, and whether they felt more confident about caring for their baby. Children’s Centre Managers and Children’s Centre Group Leaders’ questions included their expectations of the project, what worked well, memorable moments, and what they had learned.

Before any interview, the interviewer introduced themselves, explained the purpose of the interview and its link to the PCC project, and checked that the potential interviewee was still interested in taking part. The interviews were not recorded but the interviewees were informed that notes would be taken and that anonymous quotes might be used in the report and other materials. All participants gave informed verbal consent to participate. The interviewers took verbatim notes, where possible, of the key aspects of the interviews. All participants gave informed verbal consent to participate.

Parents were interviewed once. Typically, interviews with parents occurred by telephone within a week of their contact with their Parent Champion. Two parents requested the interview questions by email as they preferred this mode of contact and it fitted in better with their schedule. Most parent interviews were undertaken in the final three months of the PCC project.

Parent Champions and Core Team members were interviewed at two timepoints: timepoint 1 (at/near start of project) and timepoint 2 in the final month of the project. Parent Champions were interviewed remotely via WhatsApp (their preferred mode of contact) and Core Team members via Microsoft Teams. The Children’s Centre Managers and Group Leaders were interviewed once (by telephone or MS Teams, as per the participant’s preference), in the final two months of the project.

The handwritten notes from interviews were transcribed verbatim and analysed using thematic analysis [39]; this analysis was informed by appreciative principles. Thematic analysis followed the five stages of inductive reflexive thematic analysis (familiarisation, generating initial codes, searching for themes, reviewing themes and producing report) allowing the shift from descriptive to interpretative analysis [40]. Analysis was conducted by BC (female, PhD, academic children’s nurse), AF (female, PhD, academic children’s nurse), and JM (female, academic children’s nurse).

The following abbreviations are used with verbatim quotations: PC (Parent Champion); P (Parent); CCM (Children’s Centre Manager); GL (Children’s Centre Group Leader); and CT (Core Team). Note: the participant numbers assigned to GLs, CCMs and PCs are random and do not indicate to which Children’s Centre participants are linked. Additional illustrative quotations are presented in Appendix A.

## 3. Results

Parents (n = 43; mothers n = 42), Parent Champions (n = 9, all female), Children’s Centre Managers (n = 8, all female), Children’s Centre Group Leaders (n = 11, all female), Core Team (n = 4, n = 3 female).

Of the 61 parents who had agreed to be contacted for an interview, interviews could not be arranged with 18 parents. The reasons were as follows: did not reply to email invitations (n = 7), no reply when contacted by phone (n = 7), did not recognise the project (n = 1), disconnected the call (n = 1), wrong number (n = 2). One Children’s Centre Manager and two Children’s Centre Group Leaders did not respond to emails requesting a time to discuss the interview.

The overarching theme generated from the data was ‘Parents reaching out to parents’; this is supported by five sub-themes (see Figure 1).

### 3.1. Raising Awareness and Sharing Knowledge

The central focus of the project was on raising awareness and knowledge among parents and “helping health inequalities” (CT1), it was clear from the findings that this extended to raising awareness and knowledge among the staff at the Children’s Centres as well as into the community (e.g., nurseries and other settings). The process commenced with Parent Champions building their knowledge about bronchiolitis within their initial interactive training which prepared them for their role in disseminating the information. The initial training was a full day, face-to-face workshop supported by handouts and videos and was complemented by ongoing training at regular meetings.

#### 3.1.1. Acquiring Knowledge: An Ongoing Process and a Call to Action

Although some Parent Champions knew of bronchiolitis from personal experience with their own children, three had never heard of it. The training was received positively by the Parent Champions who noted that “right from the beginning our opinion was important and mattered” (PC2). The trainer was described as “helpful and knowledgeable… thorough… we could ask questions” (PC4) although also “felt a bit like deep water” (PC9) at times. The information was “helpful” (PC6) and “available after the session so I could refer back” (PC3) and was the right depth. The Parent Champions were interested in everything related to bronchiolitis from the “facts for Liverpool…better than just national facts” (PC8) to the statistics, risk factors, protection and prevention, signs and symptoms about what action should be taken and when a baby was suspected to have bronchiolitis. See also Appendix A. Learning was an ongoing process; most had undertaken additional research such as learning “more about the problems that poverty is causing” (PC3) as they wanted a more extensive knowledge base to support them in their role.

#### 3.1.2. Sharing Knowledge with Parents: Spreading the Word

The parents were positive about the information they received, and this was regardless of whether the information had been shared in a group or one-to-one and they said the information meant they felt more confident about looking after their baby if they had bronchiolitis. The videos showing a baby experiencing respiratory distress were identified as being memorable and helpful. The parents also appreciated the leaflets. Although all Parent Champions engaged with parents across a range of cultural backgrounds, parents whose first language was not English particularly valued engaging with Parent Champions who spoke their first language. See also Appendix A.

Parents were keen to learn about bronchiolitis. Many had not heard of it before and those who had were grateful for up-to-date and more in-depth information. They talked of how important it was to know about how to protect their baby/child and how to prevent them getting poorly. They all talked about the signs and symptoms of bronchiolitis such as “signs of the grunting and ribs—I’ll never forget that” (P15). Parents were now clearly aware of what action they needed to take, depending on the severity of their baby’s illness and they were grateful to know “about the (respiratory) red flags” (P32) as well as other things that flagged concerns “if he stops having wet nappies that’s a problem, as well” (P53).

Parents were now more aware of the support options available to them. They knew that they could seek help from their Parent Champion, dial 111 (a service provided by the National Health Service that people can dial if they have an urgent medical problem and they are not sure what to do), contact their General Practitioner (GP) or go to the drop-in centre. They were also clear that it was important to take their baby to hospital if the signs and symptoms were bad. The parents also welcomed the opportunity to ask questions about looking after their baby with bronchiolitis but also asked questions about a range of topics including breast feeding and what to do if their baby has a temperature. See also Appendix A.

#### 3.1.3. Sharing Knowledge: Spreading the Word More Widely

Parent Champions shared their knowledge with the staff within the Children’s Centres which was a “value-added” (CCM6) element, particularly since there was a lack of awareness of “how big of an issue RSV is until Parent Champion commenced in post” (CCM3).

Managers talked of how their knowledge of bronchiolitis had increased, reporting that the Parent Champion had given them “access to valuable information” (CCM4) which more widely “benefited their team” (CCM9) as staff within the Children’s Centres “are not nurses/medical” (CCM1). Group Leaders at the Children’s Centres and other stakeholders such as teachers of English as a second language valued the information. Group Leaders thought the information raised awareness about “the impact it has on A&E [Accident and Emergency] attendance and who needs hospital admission” (GL13) and that awareness was raised about “alternatives to A&E, we [GLs] learnt it’s not automatic first action, admission not always required” (GL4).

### 3.2. Creating Connection, Trust, and Confidence

#### 3.2.1. Inspiring Parents’ Trust

Parent Champions were able to connect with parents because of their friendly and approachable manner, the trust they inspired, and the way that they took parents’ concerns seriously. Parents trusted the information that Parent Champions shared as being “better or as good as information from a GP [General Practitioner] or nurse” (P7). Several parents thought that Parent Champions’ information was “more in depth than GP would be” (P10) and that their Parent Champion had “a special knowledge in the area and can see it from perspective of parents; not routine” (P3).

Parent Champions’ experiences and sensitivity meant that parents trusted them to listen, not judge and not lessen their concerns and “never made me feel stupid like I did a couple of times in the hospital” (P58). This aligned with the hopes of the Core Team who wanted the Parent Champions to “give us [acute care health professionals] a foot into the world of mums we wouldn’t see otherwise before too late” (CT3). See also Appendix A.

#### 3.2.2. Enhancing Parents’ Confidence and Informing Their Decision-Making

Parents talked about how their confidence had increased after meeting their Parent Champion not only in relation to bronchiolitis but also other aspects of their babies’ health such as “I know what to do if my baby has a high temperature and when I should call 111. I feel safe with all this information” (P51). Feeling more knowledgeable meant that rather than taking their baby straight to hospital they were confident about caring for their baby at home; “I’m more confident to go with my instinct. I know to look for things and that I can look after him at home unless he was struggling, rather than sitting in A&E for 5 h and then being sent home” (P31). Some parents were active seekers of information and previous experience with bronchiolitis had triggered them to find out, for example, “I’ve done a lot of research since he had it when he was so little, I wanted to know more. Now I know even more—it was good though to talk to [Parent Champion] (P29). See also Appendix A.

### 3.3. Flourishing in Their Role as a Parent Champion

#### 3.3.1. Fitting into the Team

Parent Champions established good working relationships and created strong connections with parents, within their own Children’s Centres and in the wider community and they valued “being part of the whole centre—people know who I am, having an NHS [National Health Service] badge, and my uniform” (PC5). This helped to develop their identity as a member of the team and parents recognised their uniform and this was perceived to increase their acceptability.

Managers talked of how well the Parent Champions fitted into their teams, for example, “engaging, helpful, flexible, good knowledge and enthusiastic, has used children centres so aware of complexity” (CCM9). Group Leaders also thought that the Parent Champions were “valued members of the team” (GL5). The Core Team were aware of how “invested they [Parent Champions] are in their role” (CT3) and that despite their own “fragile circumstances” (CT3), they have shown huge commitment as “some have left jobs to do this work” (CT2).

#### 3.3.2. Flourishing as a Parent Champion

Parent Champions brought many skills and experiences with them to their role but these skills, capacities, confidence, and sense of self as a Parent Champion strengthened across the course of the project. All talked of finding their feet, feeling more confident (especially in more challenging situations), and having a deeper knowledge of the work of the Children’s Centres, what resources were available, and how to signpost parents to help with food vouchers, support with heating, and other help.

They were passionate about their role and committed to making a change in their communities and talked about how their “job makes me smile (PC4) and that “every day is something different; so rewarding, so valuable” (PC2).

#### 3.3.3. Building Skills, Being Empathic

Their own personal experiences were key to their role, both as parents and people who had faced similar struggles; being “one of them—like a parent who has same experiences” (PC8) and with good contacts in the local community. The Core Team recognised the value of this experiential “local knowledge” (CT1). They described themselves as good talkers, being “open, [able] to talk to anyone about anything” (PC2) and good listeners and were able to “connect” (PC4), and “make people feel comfortable so they can use the information” (PC3).

There was recognition that their skills could be further enhanced through formal training and qualifications which would enhance their career prospects. One Parent Champion explained, “I’m learning all the time—every single day, I’m doing courses, new things keep happening” (PC5). One of the Core Team noted that they were “looking into the possibility of PCs doing a work based NVQ [National Vocational Qualification]. This will enable PCs to have the qualification required to apply for any vacancies that arise in centres going forward” (CT4). See also, Appendix A.

At the timepoint 2 interviews the Parent Champions were asked about what advice they would give a new Parent Champion; this basically fell into advice about being yourself, being prepared, not expecting too much to begin with, listening to parents and establishing relationships. See also Appendix A.

### 3.4. Rising to the Challenges

#### 3.4.1. Recruitment

The main challenge across the whole project related to delays and uncertainties around the recruitment process; this reflected the short time frame—six weeks—between applying for funding and the start of the project. The Core Team noted that despite “communicating well with human resources [that] working in partnership and recruiting with a new organisation was tricky…as some of the processes are complex—lots of cogs in the wheel” (CT2).

Typically, Managers wanted “improved communication, clarity re processes, dates and delays” (CCM8). The delays impacted on Parent Champions who had planned for childcare or changed other working arrangements. Other recruitment issues related to a lack of clarity about “pay, leave, expenses, long term plans” (CCM2) that managers could not answer. Recruitment remained an ongoing challenge throughout the project and some appointments were very delayed.

#### 3.4.2. Technology and Resources

An initial challenge for Parent Champions was the lack of resources such as leaflets and posters to support their work. However, they addressed this deficit and “create[d] and use[d] our own leaflets” (PC5); this was time consuming. These leaflets could not be used until they had been approved by the Project Manager. The lack of standardisation was considered both positive as well as having the potential to confuse parents moving between Centres.

Project smart phones were used for communication and showing parents videos. Tablets were ordered partway through the project to support the sharing of videos on a larger screen but were only delivered towards the end of the six-month period.

#### 3.4.3. Inclusivity

Two Parent Champions could speak a second language (Arabic, Polish) and they could deliver materials and sessions in that language; this was seen as being “a definite bonus” (GL14). Other Parent Champions used a translation app on their phones; this was seen to be reasonably effective but not something they wished to fully rely on. Greater inclusivity could have been achieved if leaflets were more widely translated “[e.g.,], into Somali, Romanian and Chinese” (GL7) and the images were more culturally relevant. Inclusivity was also fostered by ensuring that the reading age of the materials produced acknowledged that literacy levels within the target population could be low.

### 3.5. Knowledge Is Power, Prevention Is Key: The Government Needs to Know This

Parent Champions wanted people in government, policy makers and funders to know that the project worked because it was based on parents reaching out and “empowering parents” (PC2). They talked about “seeing the results” (PC4) and believed its success was based on their knowledge coming “from living day to day and living through times like the parents are experiencing first-hand” (PC3) and “start[ing] in the same place, [as] we were service users” (PC8).

The Managers, Group Leaders, and Parent Champions all shared stories about how parents had been able to make more informed decisions about their child based on their engagement with Parent Champions. This included recognising and acting upon symptoms, using more appropriate alternatives to the Emergency Department, and knowing when to take their child to the Emergency Department.

Parent Champions believed their role could be the start of fundamental change for their city as “if we can change and inform parents we can change our city—we can have healthier children living in better conditions” (PC3). They wanted the government to be aware that health promotion and prevention are not prioritised enough but are the basis for improving health; “prevention is a wonderful thing—people don’t really focus enough on this… the benefit of keeping children healthy is for everyone. It helps keep family healthy—no days off school/work this helps create a healthier society—really beneficial for everyone” (PC7). This reflected the Core Team’s ambition to improve health inequalities and their awareness that “respiratory health is part of wider geography of health—this is about everything from violence, parental well-being, poverty, smoking, and food insecurity” (CT3).

The parents wanted to raise the government’s awareness of how difficult things could be for some families, noting that “the world out there is scary for families” (PC2) living in poverty. They did not believe that people in the government would understand the desperate situation some parents are in, such as when they cannot afford to buy simple medicines for their child, noting that “a bottle of Calpol [paracetamol for children] might ‘just’ cost £3.50, but if you don’t have any money, it could be £350” (PC1), so their only alternative is go to the Emergency Department for treatment.

## 4. Discussion

The evaluation aimed to gain the perspectives and experiences of the key stakeholders about the PCC project and the difference the project made to them, and as relevant, to their family, their setting/organisation, and the community. The findings suggested that the project had positive impacts across all stakeholders and extended ‘good’ into the community. This ‘good’, coupled with the desire to promote high-value care [11] to children, is important to acknowledge when considered in relation to the high initial and ongoing costs [41] of bronchiolitis to services [7,8,10] and the socioeconomic burden on families [9].

Prior to the implementation of the project, knowledge about bronchiolitis was limited across all stakeholder groups (apart from the Core Team) and many stakeholders had either not heard of bronchiolitis or information was not contemporary. As seen in other studies, parents wanted information about bronchiolitis, including knowledge of what to look for and how to interpret the severity of symptoms [15,17] and knowing what actions to take when they became aware of symptoms and signs of bronchiolitis. Lack of knowledge about bronchiolitis has been shown to contribute to parental anxiety when their child has bronchiolitis [17]. A systematic review reported that most information is oriented to parents whose child has been hospitalised with limited information focused on parents caring for their child at home [16]; this is clearly a gap that the PCC project was trying to address.

The combination of information being delivered by a peer and the use of leaflets and videos meant that parents reported that the information was accessible, non-judgemental, and memorable. Our findings resonate with those of other studies that show parental preference for anticipatory information (that is, information prior to the child contracting bronchiolitis) [15,16,42] rather than information being provided in hospital when parents may feel overwhelmed [16]. Typically, information is provided by health care professionals [15,17], but there is not always a good match between information provided and information parents want [16,43] and no certainty that hospital-based (e.g., emergency department) health professionals routinely provide information or education [15]. Parents in our study expressed a preference for anticipatory information to be provided by a peer they could relate to, who had the time to explain things to them using language they understood, who listened to them, supported them to ask questions, and who boosted their confidence in being able to manage mild-to-moderate bronchiolitis at home. Peer-to-peer support for parents of children with long-term conditions has been shown to provide grounded reassurance and support that parents value as it comes from people who have a lived understanding of their situation [44,45,46]; this strongly resonates with the value placed on the Parent Champion appreciating the challenges they faced.

Parent Champions’ work extends beyond simple health promotion messages, e.g., ‘do this because…’. They acted as knowledge brokers about the wider factors associated with respiratory ill-health such as poverty, poor housing, and air pollution, and they acted as advocates and professional friends to parents living in their own communities. As in other studies [15,17], parents in this study were keen to learn about how to protect their baby/child from bronchiolitis and other respiratory conditions. Additionally, they were open to the Parent Champion’s sensitively shared health promotion messages. Parent Champions shared information specific to bronchiolitis but also about preventive measures such as handwashing [5] and reducing the impact of indoor pollution (e.g., avoiding the baby inhaling cooking fumes) [1], and eliminating or reducing exposure to second-hand tobacco smoke [14,24,25]. These are all modifiable risk factors [1]. This along with wider knowledge about the impact of air pollution [1,24] and deprivation [20,22,23] provided parents with a greater appreciation of the respiratory disease.

Liverpool, the city in which the intervention was implemented, has high levels of entrenched socio-economic deprivation [31]. In the absence of city-wide changes to reduce outdoor pollution, address poverty, improve housing, and reduce exposure to second-hand smoke in homes, Parent Champions via the Children’s Centres make small but important differences, one family at a time, through enhancing ‘responsive caregiving’, a core feature of nurturing care [47]. Their approach resonates with the work of community schools which have been shown to “influence the health and education of neighborhood residents though three pathways: building trust, establishing norms, and linking people to networks and services” [48].

The evaluation adopted a robust qualitative and appreciatively informed [32,33] methodology that supported the generation of 360-degree, multi-stakeholder perspectives. This ensured that the lens of interest was sufficiently broad to consider the impact on all stakeholders. The findings from the evaluation have been used during the project to inform and guide ongoing internal implementation, gain extension funding for the project, and support the implementation of the intervention in other settings in the UK.

The use of a descriptive qualitative approach results in findings that are presented in everyday rather than conceptual language, meaning they have relevance within the real world of the stakeholders. The evaluation drew on the values of appreciative inquiry meaning that the focus was on looking for ‘what worked well’ whilst also considering solutions identified to address the challenges that arose [32,33]. This affirmative approach worked well with the participants, particularly the parents, who appreciated the opportunity to positively engage with the process.

The strengths of the study include the study design, which was reported to be low burden and non-intrusive for the parents. The availability of paper-based surveys that the Parent Champions could hand out to parents, along with the availability of an e-version of the survey, was an important component. The study also had some limitations. Interview participants across all stakeholder groups were predominantly female (only one father took part in an interview). It is not possible to determine to what extent the parent participants are representative of the wider population of parents with whom the Parent Champions engaged. The findings from this study represent stakeholders in one part of the United Kingdom and the sample size is relatively small, so the findings may not be transferrable to other parts of the country.

## 5. Conclusions

The Parent Champion Project achieved its overall aim to provide peer-led support to families in the most deprived areas in Liverpool whose babies are most at risk of having severe bronchiolitis. Parent-to-parent peer support via the Parent Champions was perceived positively by the parents who welcomed the opportunity to learn and improve the lives and health of their children and the Parent Champions were successful in delivering the information.

The PCC project presents a valuable opportunity as an upstream, relatively low-cost, high-reach, innovative intervention for improving children’s respiratory health.

## Figures and Tables

**Figure 1 children-09-01479-f001:**
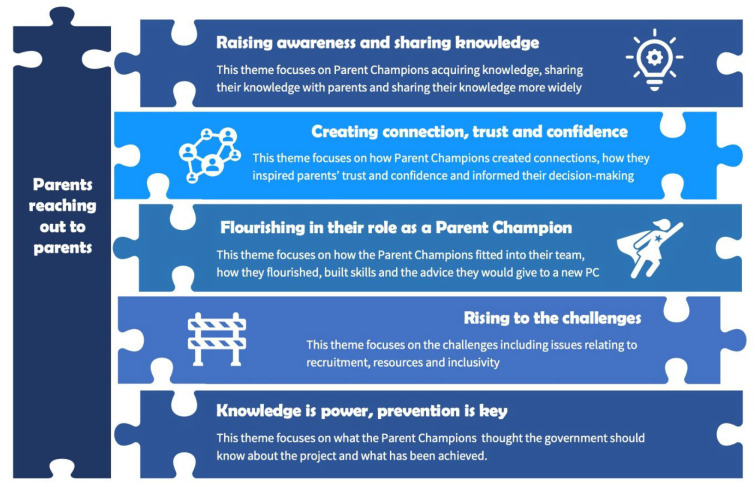
Overarching and sub-themes.

## Data Availability

The data presented in this study are available on reasonable request from the corresponding author. The data are not publicly available due to the constraints of the ethics approval.

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
