# Peer review of "Parents Reaching Out to Parents: An Appreciative, Qualitative Evaluation of Stakeholder Experiences of the Parent Champions in the Community Project"

_children, 2022, doi:10.3390/children9101479_

Round 1

Reviewer 1 Report

Dear Author

This is an article of interest to the scientific community in general and psychology in particular, but it needs to add some information to make it more robust.

The 1st question refers to the title…. If the sample is made up of only one person male and all the others are female, the title does not reflect the content of the article

2nd It is not clear how the sample was obtained, nor what the criteria for inclusion of the sample It should explain better

3rd It is not clear what the role of each group that is part of the sample is- Must explain

4th It is not clear what the role of the Line Manager is (line 98)- Should explain better

5th It is not clear how the questions were asked or their content- Must explain better

6th It is not clear how the follow-up was carried out and how the regular meetings were held- Must explain further

7th The Objective: Obtaining perspectives and experiences is not well formulated. Must reframe the objective

8th The results should be presented according to the analysis steps for a better understanding - it should reformulate

Reviewer 2 Report

I am greatful for the oppotunity to review the manuscript presented to me. I hope that the comments in the review wuld be helpful. I belive the paper is worth considering for publication, however requires minor revision. 

Comments:

- please pay more attension to using the abreviations that may be not clear enough for the world readers, e.g. GP, UK, NHS, RSV, NVQ etc.

- please explain the meaning of the number 111

- some limitation is the small size of individual groups, which may have an influence of understending the discribed problem in al country.

- please add more  references in discussion section.

- in my opinion in conclussion section should't be the citations.

Round 2

Reviewer 1 Report

Dear author, thank you for making the proposed changes which enriched your article.

with my regards